Original research

# Self-management challenges following hospital discharge for patients with multimorbidity: a longitudinal qualitative study of a motivational interviewing intervention

Carina Brandberg ®,[1] Mirjam Ekstedt ®,[1,2] Maria Flink ® [3]

[1]Department of Learning, Informatics, Managmenet and Ethics (LIME), Karolinska Institute, Stockholm, Sweden
[2]School of Health and Caring Sciences, Linnaeus University Faculty of Health and Life Sciences, Kalmar, Sweden
[3]Department of Neurobiology, Care Sciences and Society, Karolinska Institutet, Stockholm, Sweden

**Correspondence to**
Professor Mirjam Ekstedt;
mirjam.ekstedt@lnu.se

## ABSTRACT

**Objectives** The aim of this study was to describe challenges in self-management activities among people with multimorbidity during a 4-week post-discharge period.

**Design** This is a longitudinal qualitative study using data from a randomised controlled trial (RCT) of motivational interviewing (MI) sessions.

**Setting** The RCT was conducted at six wards in two hospitals—one university hospital and one general hospital in Stockholm, Sweden, during 2016–2018.

**Participants** Sixteen participants from the intervention group, diagnosed with heart failure or chronic obstructive pulmonary disease and at least one other chronic condition, were purposively selected for this study.

**Interventions** Each participant had four or five post-discharge MI sessions with a trained social worker during a period of approximately 4 weeks. The sessions were recorded digitally and analysed using content analysis. Altogether, 70 recorded sessions were analysed.

**Results** Self-management after hospital discharge was a dynamic process with several shifting features that evolved gradually over time. Patients with multimorbidity experienced two major challenges with self-management in the first 4 weeks following hospital discharge: 'Managing a system-centred care' and 'Handling the burden of living with multiple illnesses at home post-discharge'.

**Conclusions** Self-management for patients with multimorbidity in the first post-discharge period does not equate to a fixed set of tasks, but varies over the post-discharge period. Self-management challenges include not only the burden of the disease itself, but also that of navigating and understanding the healthcare system. Hence, self-management support post-discharge involves both aiding patients with care coordination and meeting their gradually shifting disease-related needs.

**Trial registration number** NCT02823795.

## Strengths and limitations of this study

► The longitudinal design with repeated sessions between patients and social workers ensure large amounts of high-quality data.

► The patients were selected, after the randomisation process, to achieve as rich a variety of data as possible, by including different genders, ages, coaches and hospital settings.

► A small number of participants (n=16); however, saturation was achieved and the repeated sessions included 70 recorded and analysed sessions.

► All patients came from same geographical area (Stockholm county, Sweden), which is a limitation regarding transferability.

one frequently used in literature—and in this study—is the coexistence of two or more chronic conditions, where all diagnoses can be of equal importance.[3,4] Healthcare systems all around the world are organised around single diseases[4–7] and most clinical practice guidelines focus on the management of a single disease.[8] Such fragmentation of services causes a large number of negative consequences, such as medical errors and misdiagnosis.[9] It is therefore not surprising that patients with multimorbidity encounter challenges in the healthcare system, or that patients are especially vulnerable during care transitions.

Discharge from hospital to home is a challenging care transition,[10] with known obstacles in the form of adverse events,[11,12] re-hospitalisation[13] and new or worsening symptoms.[14] A crucial factor for effective care transition to home is support of patient self-management, preferably initiated at the hospital; studies indicate a beneficial effect on the risk for 30-day readmission.[15,16] Self-management is 'an individual's ability to

## INTRODUCTION

In Sweden, about 25% of the population have multimorbidity, and this group accounts for 50% of the total healthcare costs.[1] There are many definitions of multimorbidity,[2] but

manage the symptoms, treatment, physical and psychosocial consequences and life style changes inherent in living with a chronic condition'.[17] A patient's capacity for self-management seems to be a crucial factor in care transitions, as high levels of patient activation have been associated with decreased healthcare usage.[18] Self-management support during care transitions is especially important, as patients often feel that the discharge encounter does not prepare them for post-hospitalisation self-management and that no one takes overall responsibility for the coordination of their care.[19] There is an international consensus that the healthcare system needs to be reorganised to provide continuity, coordination and—most importantly—a patient-centred focus that supports self-management,[19] adapted to each patient's preferences and goals.[6] People with multimorbidity are a population in true need of patient-centred self-management support, taking their preferences and needs into account.[20]

Challenges to and facilitators of self-management have been extensively explored, but self-management over time is scarcely described.[21] There are even fewer descriptions of how challenges to self-management are experienced and handled during the most vulnerable period, that is, the first weeks after a hospital discharge—to our knowledge there are none. This study aimed to describe the process of and challenges to self-management activities, as expressed by patients with multimorbidity in a 4-week post-discharge motivational interviewing (MI) consultation trial.

## MATERIALS AND METHODS
### Study design
This is a longitudinal qualitative study using inductive qualitative content analysis for analysis. The longitudinal approach was used to detect changes over time in patients' self-management challenges.[22] The data consists of descriptions of self-management challenges after hospitalisation by patients with multimorbidity, collected in four to five MI sessions per patient in a randomised controlled trial (RCT). For details of the RCT, see the study protocol.[23] A pilot study with three participants was performed before the start of the study, to test the MI method.

The Consolidated criteria for Reporting Qualitative research were used for explicit and comprehensive reporting.[24]

### The MI intervention
The MI is grounded in self-determination theory, a macro-theory of human motivation. In this study, MI was used to increase patient activation in self-management through enhancing a person's autonomy, competence and relatedness.[20] When people become more motivated, engaged and experience more psychological well-being, this predicts positive health behaviour.[25] Three specific components of autonomy support are suggested: taking the perspective of the patient, providing the patient

with choices and providing a rationale when choices are not possible.[26 27] The principle of autonomy support is at the heart of the clinical approach called MI,[28] which can be delivered in a single session or through multiple sessions.[29]

The patients in the intervention group each had four to five MI sessions with one of three trained hospital social worker during a period of approximately 4 weeks. The first session was conducted 1–2 days after discharge, to establish a first contact and capture any questions that arose early post-discharge. The following sessions were conducted 1, 2, 3 and 4 weeks post-hospitalisation. After the fourth session some patients found that they managed their situation and wanted to end the coaching. The MI sessions were held either over the phone or face-to-face in clinic, depending on what the patient preferred. The length of the MI sessions varied between 10 and 90 min. The first session was usually the shortest, due to fatigue and having a lot of information to absorb post-discharge. All sessions were audio-recorded with the permission of the patients. (The manual for the MI sessions is presented in online supplemental appendix 1).

All three hospital social workers had MI training and received MI guidance from a psychologist, a member of MINT (Motivational Interviewing Network of Trainers), once a month during the trial. The core of the MI sessions was to motivate and empower the patient to take active part in self-management regarding four important aspects and to create goals for these aspects: (1) managing medications, (2) managing symptoms and/or signs of worsening illness after discharge, (3) acquiring knowledge of follow-up and (4) acquiring knowledge of and control over whom to contact for different healthcare needs.[23] The patients were also encouraged to discuss other aspects of their self-management that were important to them. The hospital social workers had no prior relationship with the patients.

### Selection of participants and data
Recruitment for the MI intervention[23] was conducted at six wards in two hospitals: one university hospital and one general hospital in Stockholm, Sweden. Data were collected from 24 August 2016 to 17 May 2018. The participants were recruited to the RCT during hospitalisation for any cause, if they fulfilled the following inclusion criteria: diagnosed with chronic heart failure (CHF) or/ and chronic obstructive pulmonary disease (COPD), able to speak and understand enough Swedish to participate in the MI sessions, not diagnosed with dementia or mild cognitive impairment and not having a non-resuscitate statement in their medical record. Patients received both written and verbal information about the study and the authors, with the possibility to discuss any questions they might have. Those who needed time to decide were offered that until discharge.

From a total of 207 patients in the RCT, 16 patients from the intervention group (n=104) were purposively selected to cover as many aspects as possible in this

heterogeneous group of patients. Among participants with at least two chronic conditions, we sought a variation in type of diagnoses, ages, genders and hospitalisation at different hospital wards. The patients were also selected to cover MI sessions held by all three coaching social workers and throughout the intervention period, to get a variation in coaches experiences. An overview of the patients' characteristics can be found in table 1. The MI sessions, that is, conversations between the patients and the social workers, were audio-recorded and transcribed verbatim. Transcribed data consisted of four to five MI sessions per patient: 70 recorded sessions in all. When the sessions of 16 patients had been analysed, we analysed two more series of sessions (ie, from two patients) to explore if we had reached data saturation of themes for individuals.[30] As no additional individual themes emerged from this analysis, we concluded that we had reached sufficient data saturation. No participants dropped out.

## Analysis

The analysis was conducted in two phases: a qualitative content analysis of each individuals' MI sessions,[31] and a longitudinal analysis of changes in self-management process based on the content analysis.[22] Longitudinal analysis is a qualitative research approach that is well-suited to detect changes across time periods.

The transcribed sessions were analysed in several steps using inductive qualitative content analysis, including open coding and creating categories through abstraction.[31] Researchers' reflections were continuously written down during the analysis and these reflections were used in the analysis process. Initially, the full sequence of the four to five recorded MI sessions for each patient was listened to, and the transcripts were read as a whole, to achieve a comprehensive understanding of the experience of self-management after discharge. In the first phase of the analysis, units of meaning were selected, condensed and coded for one participant at a time, starting from the first session and continuing in chronological order. Thereafter, an axial analysis was conducted where codes from all patients in each session separately were abstracted and formulated to subcategories, that is, all codes from the first sessions, then all from the second sessions and so on. The session-specific subcategories were thereafter sorted into 14 subgeneric categories, and these were abstracted into two generic categories. Finally, a longitudinal analysis was conducted across the five sessions with focus on changes in the self-management process over time.[22] This longitudinal analysis identified a main category across the generic categories, 'dynamic self-management process after discharge.' An example of the analytical steps from units of meaning to generic category can be found in table 2.

CB made the first reading and coding of data. CB is a doctoral student and received continuous supervision by MF and ME during the study and obtained training within qualitative research. The coding and interpretation processes were conducted in continuous discussions

between the three authors (CB, MF and ME). Excel was used to sort codes and categories. CB and ME are registered nurses, and MF is a social worker. MF and ME are well experienced in qualitative research.

**Table 1** Characteristics of the participants (N=16).

| Characteristics | Number (SD) |
|---|---|
| Age, mean | 71 (10) |
| Male, mean | 69.3 (11) |
| Female, mean | 72.1 (9.5) |
| Gender | |
| Female | 7 |
| Place of birth | |
| In Sweden | 12 |
| Marital status | |
| Married | 7 |
| Single/widowed/separated | 9 |
| Closest next-of-kin | |
| Relative* | 14 |
| No next-of-kin | 2 |
| Education level | |
| No education | 1 |
| Elementary school | 5 |
| High school/upper secondary school | 7 |
| College/university | 3 |
| Income (Swedish krona/month) | |
| <10 000 | 1 |
| 10 000–20 000 | 6 |
| 20 000–50 000 | 7 |
| >50 000 | 1 |
| Specific diagnosis† | |
| Congestive heart failure | 9 |
| Chronic obstructive pulmonary diseases | 9 |
| Hypertension | 7 |
| Diabetes | 6 |
| Renal failure | 3 |
| Anaemia | 3 |
| Chronic conditions per individual (N) | |
| 2–4 | 2 |
| 5 | 6 |
| 6 | 4 |
| 7 | 4 |
| Diagnoses (N), males, mean | 6 (1) |
| Diagnoses (N), females, mean | 5 (1.4) |
| Charlson comorbidity index, age-adjusted, mean | 6.7 (2.5) |

*Husband/wife/partner, child or friend.
†Data taken from electronic/medical records.

**Table 2** An example of the analytical steps from units of meaning to generic category (ID 16).

| Unit of meaning | Heading (code) | Subcategory | Subgeneric category | Generic category |
|---|---|---|---|---|
| Mm…and with home care and other help, you are never really free. | Need for home care decreases sense of freedom. | Need for independence delays care seeking. | Personal experiences direct care seeking behaviour. | 'Handling the burden of living with multiple illnesses post-discharge.' |

### Patient and public involvement

Patients were not explicitly involved in the design or analysis of this study, but the study was designed based on extensive exploration of literature and own research. However, the used method (MI) in this study is patient-centred, meaning that the topics discussed during the intervention were guided by patients' needs and interests.

Patients were offered to take part of the result and those who are interested will receive the published article including a plain language summary.

### RESULT

Our data showed that patients with multimorbidity struggled with self-management in the first 4 weeks at home post-discharge. The longitudinal main category identified, 'A dynamic self-management process after discharge,' consisted of two generic categories that characterised this transitional period in a patient's chain of care: 'Managing a system-centred care' and 'Handling the burden of living with multiple illnesses post-discharge.' In the results, the longitudinal main category is first presented followed by the generic categories. The longitudinal main category does not contain any quotations, as it targets the changes in the self-management process over time.

### A dynamic self-management process after discharge

The longitudinal analysis showed that self-management after hospital discharge was a dynamic process with several shifting features that evolved gradually over time. Difficulties that appeared during the first week were replaced by new obstacles in the second or third week, that is, the challenges shifted as the patients recovered and gradually returned to normal life. During the first week post-discharge, the first two sessions concerned how the patients adjusted to their changed health conditions and the patients had many questions, especially regarding medication management. Some of the most salient challenges were fatigue, symptoms impairing activities in daily life and causing social isolation and handling of new medical regimens. The post-discharge period was also characterised by a struggle to understand the new situation, including understanding information from the hospital.

As fatigue gradually declined, a new topic of health-related anxiety and stress was raised during the second week post-discharge (ie, the third session). For example, anxiety and stress related to feelings of being a burden to relatives, neighbours or colleagues, being unsure which healthcare contact was responsible for care post-discharge, or not receiving sick pay from the Swedish Social Insurance Agency. Questions regarding management of stress-related problems dominated the following sessions and peaked during the fourth week. A reduction of fatigue and other symptoms impairing daily life could be seen at the last session.

### Managing a system-centred care

The patients described the period following hospitalisation as not being centred around their needs, but being based on the design of the healthcare system. The patient had to adapt to the system, rather than the system being aligned to their needs as a patient with multimorbidity. "As it is now, I have to chase every healthcare professional myself. It isn't the role you should have when you are on sick leave. The whole idea should be that the system takes care of you, not that you should chase the system." (ID 105)

Different physicians used different treatment strategies and several of the patients felt that new general practitioners (GPs) lacked knowledge about them and their clinical picture. "I have a cardiologist and I have a pulmonologist and I have a general practitioner then, and these three never meet." (ID 20)

This could result in insufficient information on diagnosis and treatment, when to take medicines and possible side effects of treatment. "Some doctors do not want to prescribe my medications. Medications that I should have, that I have taken for a long time." (ID 4) Patients in this study also described difficulties with getting in touch with their GPs, sometimes resulting in prolonged improper medical treatment. "… then I have to keep track of this doctor at the healthcare centre and talk to him during his phone hours, as he has 30 min 2 days a week and so he has no phone hours today." (ID 25)

High turnover of GPs, shortage of time in consultations and neglect of the patient perspective contributed to non-trusting relationships between patients and GPs. The high turnover of GPs affected patients' continuity of care. "I have been assigned to him, I have not chosen a doctor myself and everything feels very uncertain. You have to start all over again and that is really hard." (ID 51)

For patients with newly diagnosed heart failure, the follow-up after discharge was conducted at a heart failure reception, where each patient got further information, the possibility to ask questions and, if needed, evaluation and adjustment of medicines. Despite this, patients lacked an opportunity to discuss their medications with a physician, instead of having a specialist nurse evaluating the physician's prescriptions. In this way, the heart failure reception was perceived to add to the fragmentation through

involving yet another healthcare contact. "Yeah, so I met with the nurse at the cardiac ward this past Monday and I don't know, it felt a little out of sync … what the doctors said and what she said, if you can put it that way." (ID 105) In one case, a patient showed creativity by making an Excel file organising all his 21 healthcare contacts by colour, based on what diagnosis they provided care for. During the MI sessions, strategies on how to prepare for healthcare encounters to overcome the lack of continuity were addressed, for example, by discussing important topics to mention at meetings.

Another challenge was that medical record systems differed between departments, forcing patients to take responsibility for gathering information on their own care from their various caregivers. One man said (ID 50): "The healthcare system is made for only one diagnosis at a time." Several participants solved this issue by always carrying their most current patient record and list of medications with them.

Another issue was when the choice of medicine was ruled by price rather than to simplify drug management for the patient. A woman said (ID 69): "They (pharmacy) want to give me Becotide because it is cheaper at the pharmacy, but my heart does not like Becotide and my lung doctor has said 'You must not change it.' And they (pharmacy) say, 'But it is more expensive,' but I'm paying for it." For some patients, the national pharmaceutical policy of offering the cheapest products complicated drug management, as they found it difficult to understand and remember the different generic names. Another barrier regarding medicines was incomplete lists of medicines, especially in the case of temporary medicines, which are often missing from such lists.

### Handling the burden of living with multiple illnesses at home post-discharge

Lack of energy and strength in the first weeks after discharge affected the patients' ability to understand information on how to manage new symptoms, and to make the necessary adjustments in daily life. A man said (ID 36): "but I'm a patient, I'm not a normal person, I'm a patient, I'm not functioning. A man on new medication post discharge" (ID 105): "…should I sort of get more tired than this, then I will not function as a human being … I plan my activities by… doing something in the morning, because I know that around 12, I will have to go to bed and sleep 3 hours".

Fatigue and impaired physical functioning also led to the patients feeling socially isolated in their homes. For example, having to take just a few steps on stairs with impaired balance, which was a challenge already under ordinary conditions, now significantly affected daily life. In one case, a woman's fear of sudden dizziness and a risk of falling on the stairs led her to repeatedly cancel appointments with her GP. Some of the patients also expressed uncertainty and had a lot of questions when they got home. "…if you have undergone surgery or something at hospital, you receive this note and the doctor rambles on a lot but then, when you get home, you wonder what they actually said." (ID 25)

Self-management also required high level of attentiveness to subtle symptoms in order to identify diagnoses correctly and respond properly. Some patients expressed concerns regarding their ability to identify which disease their symptoms derived from. For example, a patient with COPD, CHF and anaemia (ID 20) misinterpreted her breathing problems as related to her COPD and thus acted on the symptoms 'as usual'. She said: "Yes, you get used to feeling bad". This delayed her from seeking care, which led to hospitalisation for what turned out to be serious anaemia. Some patients also expressed difficulties in identifying possible side effects when prescribed several new medicines at once. Some found their own ways of managing their daily medication, for example, by delegating this responsibility to relatives.

An acute hospital stay was experienced as a setback, which reduced strength and affected motivation for self-management. It was perceived as having to start all over again, to regain strength and independence in daily life. Keeping physical activity levels up was regarded as especially stressful after a stay in hospital. "I had a walk…or treadmill for a long time, but I haven't been able to walk on it, since I've been in such bad shape, but now… And then there's so much else going on around me and…" (ID 7) Most patients were aware of the importance of physical activity for managing daily life. Challenges to exercise included health limitations, repeated infections, surgery, lack of company and inclement weather.

The strong desire for autonomy was an asset in the handling of symptoms and challenges post-discharge. In contrast, adjusting to being dependent on home healthcare or informal care, due to increased fatigue or impaired health, was hard for some. Some participants also described having felt shame since childhood when asking for help, and taking pride in caring for themselves, which was strongly associated with their self-image. One woman said: "I want to take care of myself as well, that's the main thing I do". (ID 51) Such striving for independence could be a health risk due to seeking care too late or refusing treatment such as dialysis because of fears of becoming dependent.

In some cases, a care relationship with a relative or friend may have delayed getting adequate help. In one case, a very attentive neighbour provided food and care for a woman who could not get out of her couch after three serious falls and multiple fractures, which delayed the hospitalisation by 4 weeks. In another case, a patient was cared for at home by her husband for several weeks before she went to hospital, and only then did she realise that she had serious anaemia. Lacking the motivation to book an appointment with primary care was also related to having a poor relationship with or lacking trust in the GP; some waited for so long that they had to seek emergency care instead. "What can I say, I'm a little put off by the healthcare centre that I've been going to all these years, because they've given me Madopark (levodopa)

and antidepressants when I came to them for fatigue and chest tightness in May, so I'm a little like … Yeah, when it comes to my heart, I don't feel that I trust my GP with that anymore. If you have a cold or something, that's different. That isn't life-threatening." (ID 25)

## DISCUSSION

The result of this analysis of a series of MI sessions during 4 weeks post-hospitalisation showed that self-management after discharge is a dynamic process that is affected by managing a system-centred care and the burden of living with multiple illnesses. Overall, patients with multimorbidity need support during the first couple of weeks post-discharge, implying that they are sensitive to a system-centred healthcare system.

The period immediately after discharge from hospital is a vulnerable one, with increased risk of new hospitalisations. Patients with multimorbidity may be especially vulnerable in the management of their chronic illnesses. Our results, in line with previous research,[25 32] showed that self-management is a dynamic process that changes rapidly during the first weeks post-discharge. The patients experienced shifting needs and struggled with self-management in a system that was designed for a single-disease population, not one of increasing age and with an increasing number of diseases. Thus, the patients experienced a triple whammy, as they had to handle the burden of their illness, including fatigue, the burden of self-management of multiple diagnoses and the burden of being in a system not designed to meet their needs.[5–7 33] In this, the patients experienced that they were left alone to handle their self-management tasks, which especially affected those who were unsure about their own abilities.[34] The patients' experiences of a system that did not meet their needs could be reinforced by the design of the Swedish healthcare system, which has one of the shortest lengths of hospital stays in the European Union, EU, (5.9 days) and the third lowest percentage of general practitioners among countries in the Organisation for Economic Co-operation and Development, OECD.[35] This affects this group of patients, due to a need for prompt follow-up within primary healthcare post-discharge.

The current study also highlights that patients cannot cope with more activities than what is 'mandatory' after discharge, as they struggle with self-management while being fatigued. A large proportion of those invited to participate in the RCT in this study declined to participate due to fatigue and severe symptoms,[36] indicating that patients with complex care needs might need support for more than just self-management during the post discharge period. The system-centred healthcare system was a suboptimal condition for motivation and self-management of illness post-discharge. The population in this study was in a great need of healthcare that could provide them with a patient-centred and family-centred approach, as suggested by other studies.[3 37] Initiatives in this direction are seen within a national healthcare reform.[38] However, understanding how to provide this type of care efficiently is an enormous challenge.[3]

## Strengths and limitations

A strength of this study is the longitudinal design with four to five MI sessions, conducted during a 4-week post-discharge period. However, there is a need to consider that the audio-recorded MI sessions probably affected the post-discharge period in several ways, as they aimed to increase patients' motivation for self-management. The longitudinal main category 'dynamic self-management process after discharge' was considered to capture the patients' transition processes and their experiences of how self-management and recovery changed during the first weeks post-discharge. However, this process might have been different for persons with multimorbidity who did not receive MI sessions.

The MI sessions were directed at four aspects relevant to post-discharge self-management: (1) medication, (2) symptoms, (3) follow-up and (4) whom to contact for different healthcare needs. However, in accordance with the MI methodology, patients were also encouraged to discuss any relevant problems that occurred post-discharge, meaning that the sessions were guided by patient needs. The results indicated that the patients felt free to discuss several different aspects, both the four mentioned above and others. The MI coaches were hospital social workers, that is, not medically trained staff, which could have led to the sessions focusing on more general aspects of self-management and not specifically on medical aspects.

To ensure trustworthiness,[30] researchers' reflections were continuously written down during the analysis. These reflections were used in the analysis process. The analysis was repeatedly discussed between the three authors and within the research group. Examples of quotations from the interviews are presented in the findings. The patients were selected to achieve as rich a variety of data as possible, by including different genders, ages, coaches and hospital settings. We did not return the results to the participants for verification, as we considered that this group of old, sick patients might not be able to assess the correctness of the analysis.[30]

The data collection ensured large amounts of high-quality data, thanks to the repeated sessions between patients and social workers. These sessions contributed to building trust between patients and coaches, which probably made it more likely that rich information was obtained and that any misunderstandings or distortions were uncovered.[39]

## CONCLUSION

Self-management for patients with multimorbidity in the first weeks after hospitalisation does not equate to a fixed set of tasks, but varies over the post-discharge period. Self-management challenges include not only the burden of the disease itself, but also that of navigating

and understanding the healthcare system. Hence, self-management support in the post-discharge period involves both aiding patients with care coordination and meeting their gradually shifting disease-related needs.

**Acknowledgements** The authors thank all the patients for their participation in this study. The authors would also like to thank Marita Renman-Bengtcén, social worker, Camilla Lundström, social worker and Camilla Kjellberg, social worker, for performing all the motivational interviewing coaching sessions with the patients, and Johan Holmberg, psychologist and member of Motivational Interviewing Network of Trainers (MINT), and Sara Hammer, psychologist and member of MINT, for mentoring and providing insightful coaching to the coaches.

**Contributors** ME and MF contributed with the conception and design of the study and the obtaining of funding. CB and MF performed inclusion of all participants in the randomised controlled trial. CB conducted the initial coding of transcribed data in discussion with ME and MF and then performed a longitudinal analysis with a focus on changes in the self-management process over time. In discussions with CB, MF and ME contributed to the abstraction, interpretation of codes and formulation of subcategories and main categories. CB drafted the manuscript. MF and ME contributed with substantial contents to the writing. All authors read and approved the final manuscript prior to submission.

**Funding** Funding was received from the Doctoral School in Health Care Sciences (FiV) (#4-3084/2014), Karolinska Institutet, FORTE (#2015-00412) and Vårdalstiftelsen (#2014-0026). The support from FiV and Vårdalstiftelsen was unconditional. The data collection, design, analysis, interpretation and reporting were performed without their interference.

**Competing interests** None declared.

**Patient consent for publication** Not required.

**Provenance and peer review** Not commissioned; externally peer reviewed.

**Data availability statement** The data sets used during the current study are available from the corresponding author on reasonable request.

**ORCID iDs**
Carina Brandberg http://orcid.org/0000-0002-2040-6951
Mirjam Ekstedt http://orcid.org/0000-0002-4108-391X
Maria Flink http://orcid.org/0000-0003-0536-0024

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
