## [Reviewer comments · BMJ Open]

ARTICLE DETAILS

TITLE (PROVISIONAL)	Self-management challenges following hospital discharge for patients with multimorbidity – a longitudinal qualitative study of a motivational interviewing intervention
AUTHORS	Brandberg, Carina; Ekstedt, Mirjam; Flink, Maria

VERSION 1 – REVIEW

REVIEWER	Sasseville, Maxime Université du Québec à Chicoutimi, Health Sciences
REVIEW RETURNED	20-Apr-2021

GENERAL COMMENTS	The aim of the study was to explore challenges in self-management using a methodology of longitudinal qualitative study. Describing the self-management process of patient using a solid time conscious methodology provides valuable insight that could lead to useful understanding of the process of care for these patients and have an impact on clinical intervention. Abstact: The results are not well presented in regards of what is in the main body of the manuscript. The conclusions could be better suited to the manuscript (see the last comment on the conclusions in the main body). The introduction positions well the main concepts and definition. The verb of action in the aim is to explore, but it does not seem like the authors used an exploratory stance, but more of a descriptive stance. Methods Study design I am not well verse in longitudinal qualitative study. The data collection and analysis well position how the longitudinal aspect is used. However, the manuscript would benefit of a reference presenting or using longitudinal qualitative design and that the main text include a description on how the longitudinal aspect of the design is having an impact on relevant aspects of the method, mainly sampling, data collection and data analysis. Could you please add precision on the qualitative design used in this study? This could help classic qualitative researchers in understanding the stance that you took in the study. It appears to
--

	me that it would be a classical qualitative description. This will maybe be done with my previous comment. The motivational interviewing intervention is well presented and contextualize the results well. The sampling method for the main study is well presented, but the sub-sampling for the qualitative study is not sufficiently described. Could the authors give more details on how these patients were selected, if maximum variation sampling was used could you precise why. The Table 1. seems to indicate that you were successful in covering a wide range of characteristics. Could you add precision on how you evaluated data saturation. It is not specified anywhere that an iterative data collection and analysis process was used. Was it an a posteriori data saturation evaluation, or you went back and recruited 2 participants to ensure that no new themes emerged? Also, was it a saturation of themes for individuals or for the longitudinal process? Could you give more precision on what you mean by latent longitudinal analysis. I don't think it is usual vocabulary in qualitative research, it would help the reader if it was explained a little more with a reference if possible. CB coded all the data and MF and ME are the authors experienced in qualitative research. Are the authors saying that CB was not well experienced in qualitative research? If it is the case could the authors give precision on how CB was supported and trained? Was the entire transcript also read by the two other authors? Patient and public involvement I am not sure what the authors mean when they indicate that, because the population under study present multimorbidity and is older they are not able to participate in the research process. I would be cautious as it is reasoning that could be interpreted in disfavour of the authors in regards with the inclusion of vulnerable people in the research process in favour of better results applicability. Results: For the entire results section: I would suggest that all main themes in the results section should be supported by quotations. At the moment, the inclusion of quotations is not systematic and some interpretations are not supported. P12 I45: the interpretation made by the authors do not match was is being said by the participant. I am not sure if that comment is relevant, and you can dismiss it, but it seems like there is no more longitudinal results after the A dynamic self-management process after discharge section. It could help the reader if you could better indicate the fact that longitudinal results are presented first and results based on individual quotation are presented afterward. This could also help sustaining why there is no quotation in that section.
--	---

	Overall, the results are well written and presented, I would suggest to just complete a revision of the authors interpretations to ensure that the quotation rightly support them. Discussion P18 l47: You described the use of analytical memos in the analysis process. This was not presented in the data collection and analysis section, you should mention it there. Conclusion It seems like the conclusion is weak in light of your results and research process. The use of longitudinal qualitative data is a strength of this study and I would but that forward to support a stronger conclusion and implication final message.
--	--

REVIEWER	Keezer, M Centre de recherche du CHUM
REVIEW RETURNED	23-Apr-2021

GENERAL COMMENTS	 • Brandberg et al. report a qualitative study on the challenges facing individuals with multimorbidity following hospital discharge, based upon a series of 4-5 interviews over time for 16 individuals recruited from a parallel randomized controlled trial. • Overall this is a fine study. It provides a window into the experiences of these individuals with multimorbidity post hospital discharge. I have only a few very small comments. • The article is very well written. Every once in a while, however, there is a slight error in tense or syntax, which should be corrected (e.g. "which mean that the patients are free to express...") • I do not quite understand the sentence in the Methods section "After the fourth session, some patients expressed that they did not need any more support, why the coaching ended after four sessions." The authors may want to consider changing it.
--

VERSION 1 – AUTHOR RESPONSE

Reviewers 1 comments	Changes made	Authors answer
Abstract: The results are not well presented in regards of what is in the main body of the manuscript.	Self-management after hospital discharge was a dynamic process with several shifting features that evolved gradually over time. Patients with multimorbidity experienced two major challenges with self-management in the first four weeks following hospital discharge: 'Managing a system-centered care' and 'Handling the burden of living with multiple illnesses at home post-discharge'.	Thank you for highlighting this need. We have revised the result section in the abstract (page 2).

The conclusions could be better suited to the manuscript (see the last comment on the conclusions in the main body). Conclusion It seems like the conclusion is weak in light of your results and research process. The use of longitudinal qualitative data is a strength of this study and I would put that forward to support a stronger conclusion and implication final message.	Managing self-management post-discharge, in a population with multimorbidity, is a process that evolves gradually over time, from extensive fatigue to emergent feelings of anxiety and stress over self-management activities. To be able to meet patient needs, promptly and coordinated patient-centered primary care is highly warranted.	Thank you for the encouraging words. We have revised the conclusions in the abstract (page 2) and in the manuscript (page 20).
The verb of action in the aim is to explore, but it does not seem like the authors used an exploratory stance, but more of a descriptive stance.	The aim of this study was to describe challenges in self-management activities among people with multimorbidity during a four-week post-discharge period.	We agree and have revised the aim accordingly in the abstract and in manuscript (pages 2 and 5)
Methods Study design I am not well verse in longitudinal qualitative study. The data collection and analysis well position how the longitudinal aspect is used. However, the manuscript would benefit of a reference presenting or using longitudinal qualitative design and that the main text include a description on how the longitudinal aspect of the design is having an impact on relevant aspects of the method, mainly sampling, data collection and data analysis	This is a longitudinal qualitative study using inductive qualitative content analysis for analysis (30). The longitudinal approach was used to detect changes over time in patients' self-management challenges. The data consists of descriptions of self-management challenges after hospitalization by patients with multimorbidity, collected in four to five MI sessions per patient in a randomized controlled trial (RCT).	We agree that longitudinal analysis needs to be elaborated. We have revised this section in the method and replaced the reference (page 5)

Could you please add precision on the qualitative design used in this study? This could help classic qualitative researchers in understanding the stance that you took in the study. It appears to me that it would be a classical qualitative description. This will maybe be done with my previous comment.	Page 5: The longitudinal approach was used to detect changes over time in patients' self-management challenges. The data consists of descriptions of self-management challenges after hospitalization by patients with multimorbidity, in four to five MI sessions per patient, in a randomized controlled trial (RCT). Page 10: The analysis was conducted in two phases: a qualitative content analysis of each individuals' MI sessions (30), and a longitudinal analysis of changes in self-management process based on the content analysis (31). Longitudinal analysis is a qualitative research approach that is well-suited to detect changes across time periods.	We have added more information on the methodological approach. Pages 5 and 10.
The sampling method for the main study is well presented, but the sub-sampling for the qualitative study is not sufficiently described. Could the authors give more details on how these patients were selected, if maximum variation sampling was used could you precise why. The Table 1. seems to indicate that you were successful in covering a wide range of characteristics.	From a total of 207 patients in the RCT, 16 patients from the intervention group (n = 104) were purposively selected to cover as many aspects as possible in this heterogeneous group of patients. Among participants with at least two chronic conditions, we sought a variation in type of diagnoses, ages, genders, and hospitalization at different hospital wards. The patients were also selected to cover MI sessions held by all three coaching social workers and throughout the intervention period, to get a variation in coaches experiences.	We have revised this section to highlight how the sampling was conducted. Pages 7-8.
Could you add precision on how you evaluated data saturation. It is not specified anywhere that an iterative data collection and analysis process	When the sessions of 16 patients had been analyzed, we analyzed two more series of sessions (i.e., from two patients) to explore if we had	We have elaborated on data saturation to make it clearer. Page 8.

was used. Was it an a posteriori data saturation evaluation, or you went back and recruited 2 participants to ensure that no new themes emerged? Also, was it a saturation of themes for individuals or for the longitudinal process?	reached data saturation of themes for individuals (29). As no additional individual themes emerged from this analysis, we concluded that we had reached sufficient data saturation.	
Could you give more precision on what you mean by latent longitudinal analysis. I don't think it is usual vocabulary in qualitative research, it would help the reader if it was explained a little more with a reference if possible.	In the first phase of the analysis, units of meaning were selected, condensed, and coded for one participant at a time, starting from the first session and continuing in chronological order. Thereafter, an axial analysis was conducted where codes from all patients in each session separately were abstracted and formulated to sub-categories, i.e., all codes from the first sessions, then all from the second sessions, and so on. The session-specific sub-categories were thereafter sorted into 14 sub-generic categories, and these were abstracted into two generic categories. Finally, a longitudinal analysis was conducted across the five sessions with focus on changes in the self-management process over time (31). This longitudinal analysis identified a main category across the generic categories, 'dynamic self-management process after discharge.'	Thank you for highlighting this, we have removed "latent" to avoid misunderstanding, and revised the section to clarify the analysis. Page 10.
CB coded all the data and MF and ME are the authors experienced in qualitative research. Are the authors saying that CB was not well experienced in qualitative research? If it is the case could the authors give precision on	CB is doctoral student and received continuous supervision by MF and ME during the study and obtained training within qualitative research.	We have added information on CB's experience, supervision and training in qualitative methods. Page 11.

how CB was supported and trained? Was the entire transcript also read by the two other authors?		
Patient and public involvement I am not sure what the authors mean when they indicate that, because the population under study present multimorbidity and is older they are not able to participate in the research process. I would be cautious as it is reasoning that could be interpreted in disfavour of the authors in regards with the inclusion of vulnerable people in the research process in favour of better results applicability.	Patients were not explicitly involved in the design or analysis of this study, but the study was designed based on extensive exploration of literature and own research. However, the used method (MI) in this study is patient-centred, meaning that the topics discussed during the intervention were initiated by patients and was influenced by their needs and interests. Patients were offered to take part of the result and those who are interested will receive the published article including a plain language summary.	Thank you for highlighting that our intention could be misunderstood. We have revised this paragraph. Page 11.
Results: For the entire results section: I would suggest that all main themes in the results section should be supported by quotations. At the moment, the inclusion of quotations is not systematic and some interpretations are not supported. P12 I45: the interpretation made by the authors do not match was is being said by the participant.		We have revised and added quotes in the result section.
I am not sure if that comment is relevant, and you can dismiss it, but it seems like there is no more longitudinal	The longitudinal main category identified, 'A dynamic self-management process after discharge,' consisted of two	The comment is relevant, and you are correct. We have

results after the A dynamic self-management process after discharge section. It could help the reader if you could better indicate the fact that longitudinal results are presented first and results based on individual quotation are presented afterward. This could also help sustaining why there is no quotation in that section.	generic categories that characterized this transitional period in a patient's chain of care: 'Managing a system-centered care' and 'Handling the burden of living with multiple illnesses post-discharge.' In the results, the longitudinal main category is first presented followed by the generic categories. The longitudinal main category does not contain any quotations, as it targets the changes in the self-management process over time.	added information to make this clearer. Page 12.
Overall, the results are well written and presented, I would suggest to just complete a revision of the authors interpretations to ensure that the quotation rightly support them.		Thank you for this positive feedback. We have added quotations so that all paragraphs are supported by at least one quote.
Discussion P18 I47: You described the use of analytical memos in the analysis process. This was not presented in the data collection and analysis section, you should mention it there.	Researchers' reflections were continuously written down during the analysis and these reflections were used in the analysis process.	Thank you for noticing this lack of clarity. We have added information on the use of memos. Page 10.
Conclusion It seems like the conclusion is weak in light of your results and research process. The use of longitudinal qualitative data is a strength of this study and I would but that forward to support a stronger conclusion and implication final message.	Self-management for patients with multimorbidity in the first weeks after hospitalization is not a given set of tasks but varies over the post-discharge period. Self-management challenges include not only the burden of the diseases but also the burden of managing the healthcare system. Hence, self-management support in the post-discharge period includes supporting patients with care coordination and meeting patients' gradually	Thank you for encouraging us to provide a stronger conclusion. We have revised this paragraph. Page 20.

	shifting needs related to the diseases.	
Reviewer 2 The article is very well written. Every once in a while, however, there is a slight error in tense or syntax, which should be corrected (e.g. “which mean that the patients are free to express...”	However, the used method (MI) in this study is patient centred, meaning that all topics discussed during the intervention were initiated by patients and was influenced by their needs and interests.	Thank you for noticing these errors. We hope that we have corrected them all.
I do not quite understand the sentence in the Methods section “After the fourth session, some patients expressed that they did not need any more support, why the coaching ended after four sessions.” The authors may want to consider changing it.	After the fourth session some patients found that they managed their situation and wanted to end the coaching.	Thank you, we have revised this sentence. Page 6.

VERSION 2 – REVIEW

REVIEWER	Sasseville, Maxime Université du Québec à Chicoutimi, Health Sciences
REVIEW RETURNED	01-Jul-2021

GENERAL COMMENTS	I reviewed a previous version of this manuscript, and the authors have thoroughly addressed all the concerns with openness and precision. I would only have a last small comment where I as a reader I think I would like to see reference 31 on longitudinal qualitative analysis appear in the study design section where it is first mention. I let the authors decide if this minor adjustment is relevant for them. Congratulations to the authors and especially to the doctoral student for the interesting work.
---

VERSION 2 – AUTHOR RESPONSE

Reviewer 1

I would only have a last small comment where I as a reader I think I would like to see reference 31 on longitudinal qualitative analysis appear in the study design section where it is first mention. I let the authors decide if this minor adjustment is relevant for them.

Our answer: Thank you Dr Sasseville for your thorough review and kind feedback on our efforts to fulfil the required changes. We believe that it contributed to a substantial improvement of the article. We agree with you that the reference 31 (Saldana) should appear in the method section where we first mention the longitudinal analysis. This is now changed, and the reference got the number 22 instead. All the subsequent references are changed accordingly. I hope you accept the revision although it is not marked with “track changes” in the marked copy.